# Control of zeolite framework flexibility for ultra-selective carbon dioxide separation

Peng Du[1,5], Yuting Zhang [1,5], Xuerui Wang[1,5], Stefano Canossa[2], Zhou Hong[3], Gwilherm Nénert[4], Wanqin Jin [1] & Xuehong Gu [1✉]

Molecular sieving membranes with uniform pore size are highly desired for carbon dioxide separation. All-silica zeolite membranes feature well-defined micropores, but the size-exclusion effect is significantly compromised by the non-selective macro-pores generated during detemplation. Here we propose a template modulated crystal transition (TMCT) approach to tune the flexibility of Decadodecasil 3 R (DD3R) zeolite to prepare ultra-selective membranes for $CO_2/CH_4$ separation. An instantaneous overheating is applied to synchronize the template decomposition with the structure relaxation. The organic template molecules are transitionally converted to tight carbon species by the one-minute overheating at 700 °C, which are facilely burnt out by a following moderate thermal treatment. The resulting membranes exhibit $CO_2/CH_4$ selectivity of 157~1,172 and $CO_2$ permeance of $(890~1,540) \times 10^{-10}$ mol m$^{-2}$ s$^{-1}$ Pa$^{-1}$. The $CO_2$ flux and $CO_2/CH_4$ mixture selectivity reach 3.6 Nm$^3$ m$^{-2}$ h$^{-1}$ and 43 even at feed pressure up to 31 bar. Such strategy could pave the way of all-silica zeolite membranes to practical applications.

[1] State Key Laboratory of Materials-Oriented Chemical Engineering, College of Chemical Engineering, Nanjing Tech University, Nanjing 211816, P. R. China. [2] EMAT, University of Antwerp, 2020 Antwerp, Belgium. [3] Nanjing Membrane Materials Industrial Technology Research Institute Co., Ltd., Nanjing 211808, P. R. China. [4] Malvern Panalytical B. V., Almelo 7600 AA, The Netherlands. [5]These authors contributed equally: Peng Du, Yuting Zhang, Xuerui Wang. ✉email: xhgu@njtech.edu.cn

Natural gas is clean energy resource and feedstock for commodity chemicals and liquid fuels[1,2]. Membrane separation and amine absorption are commercially used for $CO_2$ removal from natural gas to increase the combustion heat and to reduce the corrosive potential for pipeline delivery[3]. Membrane separation based on $CO_2$-permselective membranes is one of the most energy-efficient alternatives but it occupies only a small share (<10%) in the separation market of natural gas because of the limited performance of commercial membranes[3,4]. Zeolite membranes, such as all-silica 8-member-ring (8MR) zeolite membranes, feature uniform micropores that afford superior permeation flux and selectivity. The robust nature of mechanical strength, thermochemical and hydrothermal stability guarantee zeolite membranes long life-time under harsh conditions, such as high pressure/temperature and water vapor[5–8]. One of the main barriers to scale-up zeolite membranes for gas separation is the nonselective macropores generated during high-temperature calcination required for detemplation[9,10]. The temperature was generally up to 550~700 °C to essentially remove the template from all-silica zeolite membranes so that limited progress on the scale-up was achieved[11,12].

The mismatched zeolite framework flexibility and ceramic support expansion is deemed as the main issue to generate non-selective macro-pores during detemplation[13]. To overcome the barrier, Tsapatsis et al. developed a rapid thermal processing method to strengthen grain bonding within 10MR MFI zeolite membranes[14]. Ozonization at moderate temperature was also effective to mitigate the mismatch as proven by Kapteijn and co-workers[15]. However, the diffusivity of oxygen/ozone reactants and decomposed products (e.g., $H_2O$, $CO_2$, CO, $NH_3$, $C_2H_4$, and $C_3H_6$) in 8MR zeolite was two orders of magnitude lower than that in 10MR zeolite[16]. More than 80 h was required to achieve thorough detemplation for 8MR DD3R zeolite membranes[12], causing low efficiency and high fabrication cost. It is still challenge to develop time-efficient and facile detemplation method to produce ultra-selective 8MR DD3R zeolite membranes[17].

Herein, we report a template modulated crystal transition (TMCT) method to acquire ultra-selective DD3R zeolite membranes. The fresh DD3R zeolite membranes were initially overheated for only one minute, wherein the organic template was temporarily converted to $sp^2$ carbon species, releasing more space for structure relaxation (Fig. 1a, b). Our method synchronized the organic template decomposition with the zeolite phase transition. The less occupied zeolite cage allowed accessibility of oxidant (oxygen) and diffusivity of oxidation products. As a result, the residual $sp^2$ carbon species burned out at a moderate temperature of 550 °C, mitigating the mismatched thermal expansion between zeolite crystals and ceramic support (Fig. 1c, d). Ultra-selective hollow fiber DD3R zeolite membranes were scaled up, which are appealing to $CO_2$-rich natural gas upgrading.

## Results and Discussion

**Membrane preparation and TMCT treatment**. The DD3R zeolite membranes were prepared on four-channel $\alpha$-$Al_2O_3$ hollow fibers (Supplementary Figs. 1, 2). Non-selective macro-pores were always prevalent in the membranes detemplated by conventional calcination (CC) regardless of 550 °C or 700 °C with heating rate of 0.5 °C min$^{-1}$ (Supplementary Fig. 3c, d). Remarkably the membranes kept intact by an instantaneous overheating at 700 °C for one minute (TMCT, Supplementary Fig. 3e, f). The nonselective macropores can be visualized by confocal laser scanning microscopy (also called fluorescence confocal optical microscopy[18]) using fluorescein sodium (~1 nm) as the probe molecule, which can exclusively penetrate through and retain in the nonzeolitic macropores (Supplementary Fig. 4).

A series of micrographs were recorded on each thin slice by extending the depth away from the membrane surface and used to 3D visualization of the fluorescence emission for quantification assessment. Conspicuous fluorescence emission was observed from the CC membrane, confirming the presence of non-selective macro-pores (Supplementary Fig. 5a). On the contrary, no fluorescence emission was detected from our TMCT membranes (Supplementary Fig. 5b), indicating a perfect rejection to the probe molecule. The 2D micrographs at the depth of 3 μm are shown via semiquantitative analysis (Supplementary Fig. 5c–e). A wide span of the emission spots (~10 μm) was recorded in CC membrane, which was associated with the formation of non-selective macro-pores during the template decomposition. However, the emission intensity was too weak to be detected in the TMCT membranes.

To explore the function of TMCT treatment, DD3R zeolite powders were used to evaluate the porosity and pore accessibility. The BET area and micropore volume of TMCT sample were merely 71 m$^2$ g$^{-1}$ and 0.02 cm$^3$ g$^{-1}$ (Fig. 2a, Supplementary Figs. 6, 7). This is consistent with TGA results, wherein still 58% template occupied the zeolite cages (Supplementary Fig. 8). However, the values reached to 349 m$^2$ g$^{-1}$ and 0.14 cm$^3$ g$^{-1}$ after the following conventional calcination (TMCT + CC, Supplementary Figs. 6, 7), identical to that of typical DD3R zeolite[19]. The results reveal the template molecules were partially decomposed by TMCT treatment and completely burned out after the conventional calcination. However, the BET area and micropore volume were 25% lower if only conventional calcination was adopted, indicating that minor residual species blocked the zeolite aperture and yielded "dead volume".

To identify the generated carbon species, $^{13}C$ MAS solid-state NMR spectra and X-ray photoelectron spectra (XPS) were collected (Supplementary Figs. 9, 10) Four characteristic resonances at 29.4, 36.6, 41.4, and 46.6 ppm were observed from the fresh DD3R zeolite, which are assigned to C(4), C(3), C(2), C(1) as labelled in the 1-adamantylamine molecule (Fig. 2b)[20]. After TMCT treatment at 700 °C, these four peaks disappeared together with the appearance of two slightly separated peaks at 124 ppm and 127 ppm, assigned to $sp^2$ carbon species generated through dehydrogenation reaction. One more peak at 164.2 ppm indicates the formation of carbonyl segments, which can be further proved by the new peak at 288.9 eV in the C 1s XPS spectra (Fig. 2c). Meanwhile, nitration of template molecules occurred, shifting the peaks to higher binding energy (Fig. 2d)[21]. The zeolite cages became less occupied after the TMCT treatment, which further improved the accessibility of oxidant (oxygen) and diffusivity of oxidation products. Following this line, the template molecules trapped in DD3R zeolite were facilely removed, evidenced by the disappearance of nitrogen signal from XPS spectra.

**Structure refinement of single crystals**. To investigate the zeolite phase transition, we synthesized single crystals with size up to 109 μm for high-resolution synchrotron single-crystal XRD characterization (Fig. 3a and Supplementary Fig. 11). The significant limitation in conclusively identifying the thermal expansion of DDR-type zeolite has been its limited crystallinity, which made single crystals suitable for X-ray diffraction unavailable before[22,23]. The fresh zeolite was described by R-3m space group as the previous assignment[24], however, R-3 symmetry combined with the systematic presence of merohedral twinning was more accurate for TMCT sample (Fig. 3b, Supplementary Figs. 12–16). This allowed to unravel a staggered conformation of the silica tetrahedra oxygens along the c-axis, which deviated from the eclipsed arrangement in R-3m models. The rigid 1-adamantanamine molecules strongly braced and interacted with the framework, evidenced by the large Atomic Displacement Parameters (ADPs) of oxygen

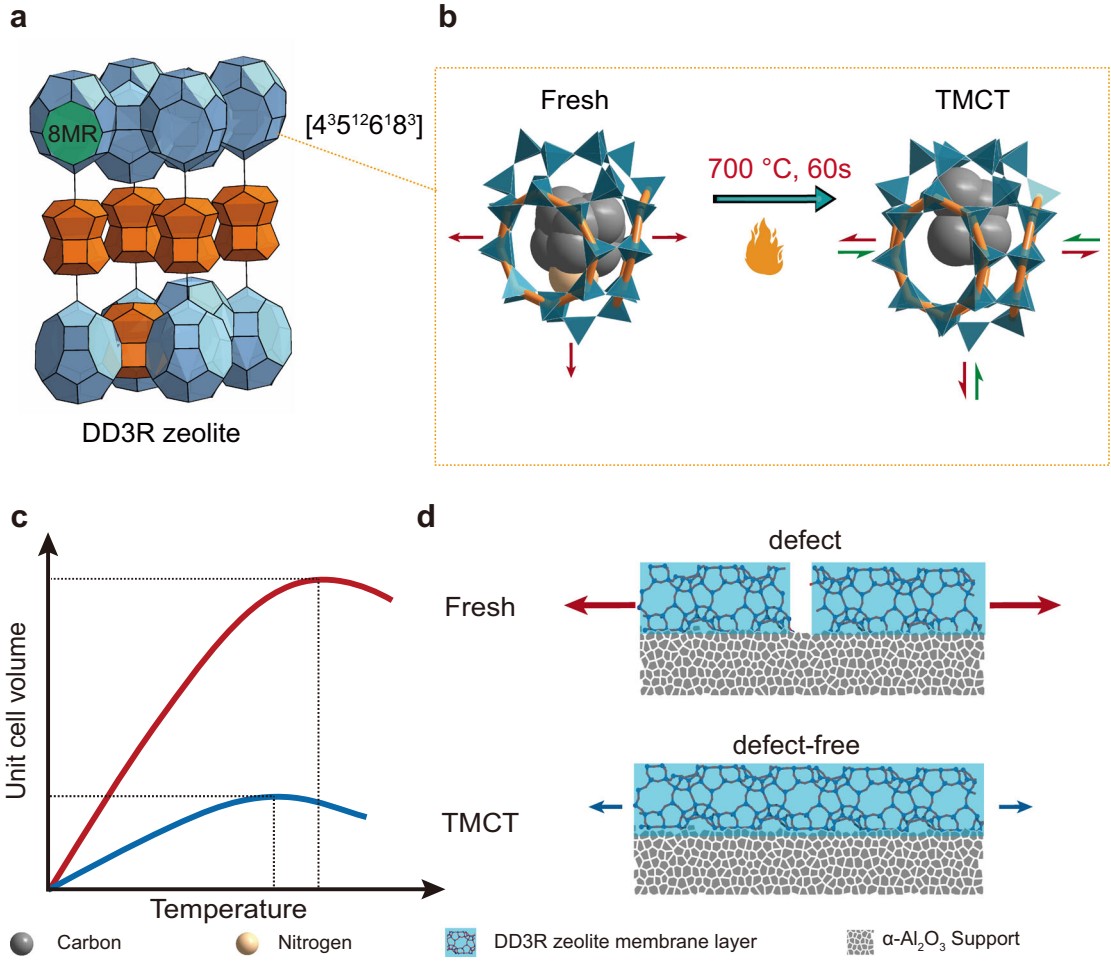

**Fig. 1 Rational design of template modulated crystal transition (TMCT). a** DD3R framework structure. The blue and orange cages denote the $[4^35^{12}6^18^3]$ and $[4^35^66^1]$ cages, respectively. The green area denotes the 8MR window. To clearly observe the structure, the $[5^{12}]$ cages were not displayed. **b** Illustration of $[4^35^{12}6^18^3]$ cage flexibility occupied by template molecule (fresh) and carbons (TMCT). The orange rings represent the 8MR window of DD3R zeolite. **c** Illustration of expected temperature-dependent expansion of fresh (red line) and TMCT (blue line) DD3R zeolite. **d** Illustration of defective and defect-free DD3R zeolite membranes after the conventional calcination at 550 °C. Fresh, as-synthesized zeolite; TMCT, overheating at 700 °C for 1 min.

atoms (Supplementary Fig. 17). After TMCT treatment, the framework was allowed to relax and transformed to R-3 symmetry. This situation was applied to the empty DDR zeolite (TMCT + CC, Supplementary Figs. 18, 19).

This is consistent with the broadening $^{13}$C-NMR peak of C3 at 38 ppm (Fig. 2b)[25]. Diffuse scattering halos centered on Bragg positions suggest the presence of correlated lattice distortions in all three samples due to the lattice continuity between the different sub-domains involved in the merohedral twinning. Moreover, diffraction patterns of TMCT and TMCT + CC crystals showed additional diffuse intensities lying between the Bragg reflections, consistent with local violations of the R-3 symmetry due to the distortion of the silica lattice upon template degradation and release. These scattering features and the decrease in ADPs after TMCT treatment was attributed to the transformation of bulky 1-adamantanamine molecule to tight $sp^2$ carbon species, followed by structure relaxation achieved via corner-sharing $SiO_4$-tetrahedra rotation about the c-axis (Fig. 3b). Most importantly, this transformation is accompanied by an overall increase of unit cell volume by ~0.45% (Supplementary Table 1).

**Mechanism of TMCT method**. The thermal expansion was further tracked by in-situ X-ray diffraction, wherein the DD3R zeolite

powders were subject to heating from room temperature to 900 °C step by step with 5 min equilibrium prior to data collection. The diffraction would be irrelevant with the ramp rate and duration time prior to 510 °C since the templates were intact at the temperatures (Supplementary Fig. 8). For fresh DD3R zeolite, the peaks at 26.2°, 26.5°, 26.8° shifted to lower angle with heating up and reached to minimum at 425 °C then increased with further heating up (Fig. 4a). To quantify the thermal expansion, the unit cell volume was determined and maximized at 425 °C (Fig. 4b). However, the temperature required to reach the maximum unit cell volume decreased to 325 °C for TMCT sample and 275 °C for empty sample (Fig. 4b and Supplementary Figs. 22–24). Most importantly, both TMCT and empty samples showed less thermal expansion, only 45.9% of the fresh one (see thermal expansion coefficients from Supplementary Table 2). The lattice parameter of a/b axis was consistent before 425 °C for the fresh sample (Fig. 4c), whereas the c axis always expanded up to 475 °C (Fig. 4d). The rigid template molecules braced the zeolite cages and weakened the transverse amplitude of the O atom[26], consequently the crystals expand until the template decomposed. While the zeolite framework was more flexible once the bulky 1-adamantanamine molecules decomposed. The tight $sp^2$ carbon generated more space in the $[4^35^{12}6^18^3]$ cavity, favoring the structure relaxation via corner-sharing $SiO_4$ tetrahedron rotation (Fig. 4e, f).

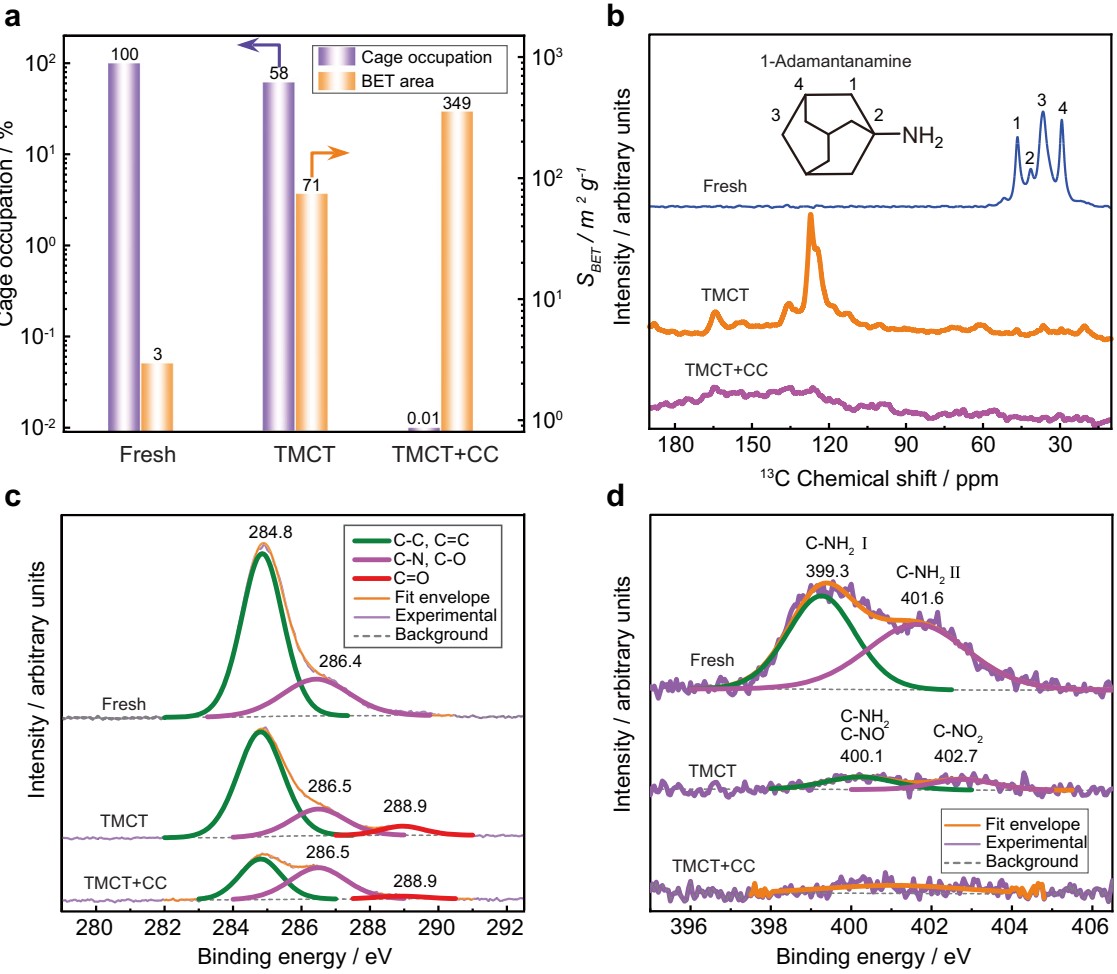

**Fig. 2 Template decomposition process in TMCT approach. a** BET area and cage occupation in DD3R zeolite. **b** $^{13}$C NMR spectra; **c** C 1s XPS spectra; **d** N 1s XPS spectra. Fresh, as-synthesized zeolite; TMCT, overheating at 700 °C for 1 min; TMCT + CC, 700 °C for 1 min followed by 550 °C calcination for 10 h. Source data are provided as a Source Data file.

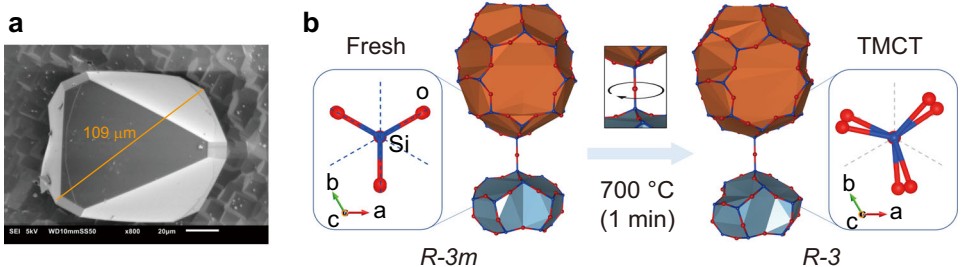

**Fig. 3 Structural relaxation of DDR-type zeolite during TMCT treatment. a** SEM image of DDR-type zeolite crystal. **b** Rotation of silica tetrahedra during TMCT procedure, transferring the space group from $R$-3$m$ to $R$-3. The silicon and oxygen atoms are highlighted in blue and red color to demonstrate the structure relaxation. Fresh, as-synthesized zeolite; TMCT, overheating at 700 °C for 1 min.

To track the nonselective macropore generation, the $N_2$ permeation of fresh and TMCT membranes were recorded during the conventional calcination in air atmosphere (Fig. 4g). The $N_2$ permeance of fresh membrane started and increased rapidly from 475 °C, which matched the temperature required for the maximized crystal expansion along $c$ direction. On the contrary, the increase was less remarkable for TMCT membrane and the eventual permeance was two orders of magnitude lower. The results indicate the non-selective macro-pores generated prior to template decomposition at 510 °C (Supplementary Fig. 8). The poor $CO_2/CH_4$ selectivity ($\alpha = 3.1$, Fig. 4h) is another

evidence on the presence of macro-pores. The framework of TMCT membrane was less limited because of the partially decomposition of template, leading to $CO_2/CH_4$ selectivity of 328.

**TMCT optimization and $CO_2/CH_4$ separation.** The TMCT procedure was further optimized in terms of temperature and time. All the fresh DD3R membranes showed an extremely low $CO_2$ permeance less than $0.1 \times 10^{-10}$ mol m$^{-2}$ s$^{-1}$ Pa$^{-1}$ for binary $CO_2/CH_4$ mixture since they were to be of high quality and their cavities were occupied with templates (Fig. 5a). After one-minute

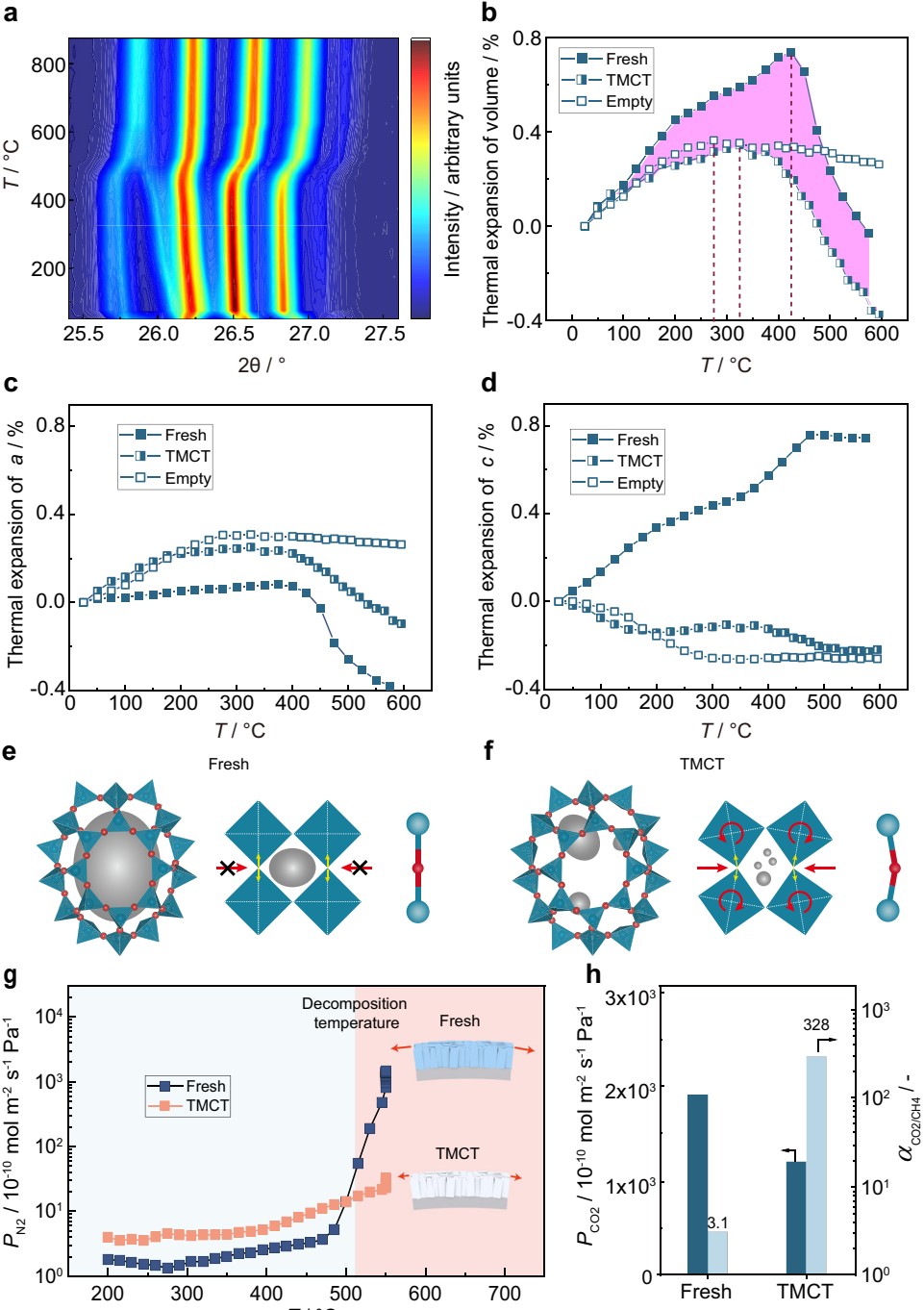

**Fig. 4 In-situ characterization by high-temperature X-ray diffraction and gas permeation. a** Isoline plot of the temperature evolution during in-situ template removal from fresh DD3R zeolite. **b** Thermal expansion of zeolite crystal volume. The pink area showed the difference in thermal expansion between fresh and TMCT samples. **c, d** Thermal expansion of zeolite lattice parameters. Fresh, as-synthesized zeolite; TMCT, overheating at 700 °C for 1 min; empty, 700 °C for 1 min followed by 550 °C calcination for 10 h. **e, f** Schematic diagram of rigid unit modes (RUMs) in zeolite structure. Red and blue nodes represent O and Si atoms, respectively. Gray blocks represent the guest molecule in the cage. **g** Pure $N_2$ permeance of fresh and TMCT membrane as function of calcination temperature with a heating rate of 0.5 °C min$^{-1}$ in air. **h** Binary $CO_2/CH_4$ separation performance of the membranes from (**g**). Test condition: room temperature and 2 bar feed pressure. Source data are provided as a Source Data file.

overheating (TMCT) treatment, $CO_2$ permeance increased from $1.5 \times 10^{-10}$ to $22.4 \times 10^{-10}$ mol m$^{-2}$ s$^{-1}$ Pa$^{-1}$ as the temperature elevated from 600 °C to 800 °C, which can be explained by the formation of less bulky $sp^2$ carbon species and the facilitated dehydrogenation of template at higher temperature[27]. After the following moderate calcination, a maximum $CO_2/CH_4$ selectivity

of 442 was achieved at $T_{TMCT} = 700$ °C, while the selectivity was 109 at $T_{TMCT} = 600$ °C and 55 at $T_{TMCT} = 800$ °C respectively.

When the TMCT time was prolonged from 1 to 8 min, the resulting $CO_2$ permeance increased from $760 \times 10^{-10}$ mol m$^{-2}$ s$^{-1}$ Pa$^{-1}$ to $890 \times 10^{-10}$ mol m$^{-2}$ s$^{-1}$ Pa$^{-1}$ whereas the $CO_2/CH_4$ separation selectivity gradually fell down from 442 to 165 after CC

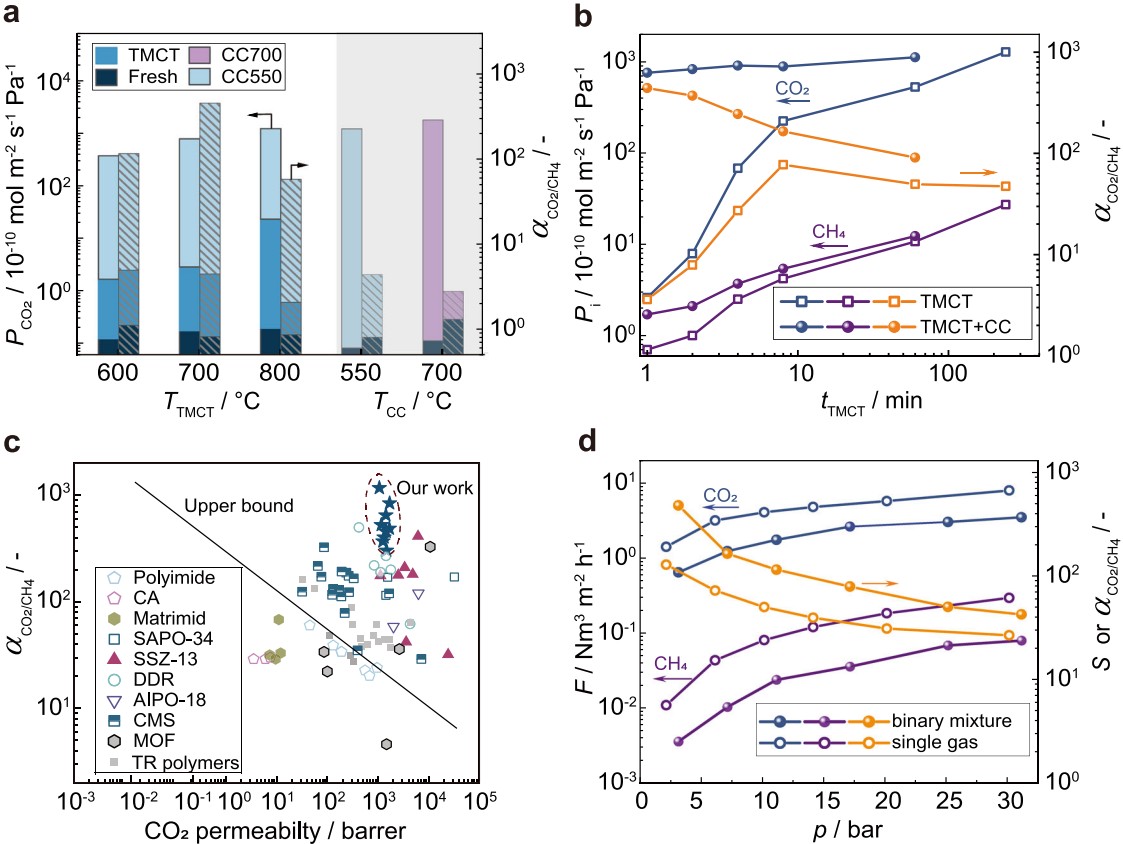

**Fig. 5 TMCT procedure optimization and CO₂/CH₄ separation performance. a** Investigation on TMCT (one-minute overheating) temperature and $CO_2$/ $CH_4$ separation performance comparison with the membranes detempled by conventional calcination at 550 °C (CC550) and 700 °C (CC700). **b** Effect of TMCT period on $CO_2$/$CH_4$ separation performance. **c** Comparison on $CO_2$/$CH_4$ separation performance, involving commercial cellulose acetate (CA) membranes[31], polyimide membranes[29], Matrimid membranes[30], carbon molecular sieve (CMS) membranes[35–37], MOF membranes[38–40], and other 8MR zeolite membranes[28,32–34,52]. **d** Gas permeation flux ($F$) and $CO_2$/$CH_4$ selectivity of 20 cm hollow fiber DD3R zeolite membrane in pure components (open symbols) and equimolar mixture (solid symbols) with pressure range of 3 to 31 bar (condition: room temperature, without sweep gas). Source data are provided as a Source Data file.

(Fig. 5b), implying that more and more non-selective macro-pores generated with TMCT time extension. To identify the period of macro-pore formation, both $CO_2$ and $CH_4$ permeances were recorded on the TMCT membranes. The $CO_2$/$CH_4$ selectivity initially increased and reached the maximum value of 77 after 8 min TMCT treatment, much higher than that of CC700 membrane ($\alpha = 1.5$). The results prove the overheating with 700 °C min⁻¹ created less nonselective macropores. While the selectivity decreased to 47 when the TMCT time extended to 240 min. The thermal expansion of alumina support essentially exceeded the zeolite layer at 700 °C. That is the reason why a short overheating period (<8 min) and a moderate temperature of 550 °C was subsequently applied to burn out the residual template (Supplementary Fig. 26).

In order the verify the scalability, large-area hollow fiber DD3R zeolite membranes were prepared and detempled by our TMCT strategy. We conducted one-pot synthesis of 17 pieces with total membrane area of 405 cm² and each membrane has an effective length of 20 cm (Supplementary Fig. 27). The $CO_2$/$CH_4$ selectivity ranged from 157 to 1,172, giving an average value of 422 (Supplementary Table 3). The TMCT approach was also feasible to MFI and SSZ-13 zeolite membranes (Supplementary Table 4). The $CO_2$/$CH_4$ separation performance of DD3R zeolite membranes obtained in this work far surpassed the well-known "upper bound" defined by polymeric membranes[28] (e.g., polyimide[29], Matrimid[30], and the commercial cellulose acetate[31], Fig. 5c). Our DD3R zeolite membranes were the most $CO_2$-selective compared to other zeolite membranes (e.g., SAPO-34[32], SSZ-13[33], AIPO-

18[34]), carbon molecular sieve membranes[35–37], and MOF membranes (e.g., ZIF-8[38], MOF-1[39], ZIF-62[40]) in spite of their moderate permeabilities.

Natural gas is always fed at high pressure[3], but such condition is rarely concerned in studies on $CO_2$/$CH_4$ separation through zeolite membranes[12,41]. The membrane was tested in single gas permeation and $CO_2$/$CH_4$ binary separation under pressure up to 31 bar. High $CO_2$ flux was favored at high feed pressure due to the increased adsorption loading (Fig. 5d and Supplementary Fig. 28), which enhanced the driving force for gas permeation. When the feed pressure was up to 31 bar, the $CO_2$ flux was 8.1 Nm³ m⁻² h⁻¹ in pure feed gas and 3.6 Nm³ m⁻² h⁻¹ for binary mixture with a mixture selectivity of 43. The permeate $CO_2$ concentration reached 97.7% only in one-stage membrane separation. The membrane was further used to quantify $CO_2$ and $CH_4$ permeation through zeolitic pores and defects. Zeolitic pores contributed 99.99% $CO_2$ permeance even under 31 bar (Supplementary Fig. 30), confirming the membranes were to be of high quality. The slight decrease in $CO_2$ permeance through zeolite pores under high feed pressure was caused by monotonic reduction in $CO_2$ diffusivity dependent on its nonlinear adsorption loading[42]. For $CH_4$, its diffusion in zeolite pore was dominated by the energy barrier that molecules hopping through 8MR windows between cages. The additional $CH_4$ loading reduced the net energy barrier, accelerating the $CH_4$ diffusivity[43]. Thus, $CH_4$ permeance in zeolite cages increased with the feed pressure. Meanwhile $CO_2$/$CH_4$ selectivity with different feed compositions ranged from 39 to 50

at feed pressure of 31 bar (Supplementary Fig. 32). It can be anticipated that the hollow fiber DD3R zeolite membranes would be of great interest in $CO_2$-rich natural gas upgrading owing to their virtue of high selectivity.

In summary, we developed a template modulated crystal transition (TMCT) approach to fabricate ultra-selective DD3R zeolite membranes in hollow fiber geometry. The framework flexibility was well controlled by the rapid template decomposition. The TMCT treatment ensured a less thermal expansion and an improved diffusion of oxidant, benefitting the template removal at a moderate temperature. 17 pieces of large-area membranes were obtained in one-batch synthesis, wherein an unprecedented $CO_2/CH_4$ selectivity of 1172, as well as a $CO_2$ permeance of $890 \times 10^{-10}$ mol m$^{-2}$ s$^{-1}$ Pa$^{-1}$ was achieved. The membrane performance far surpassed current commercial polymeric membranes and the upper-bound. The TMCT approach overcame the barriers for the preparation of ultra-selective DD3R zeolite membrane and paves the way to $CO_2$-rich natural gas upgrading based on the zeolite membranes.

## Methods

**α-Al$_2$O$_3$ hollow fiber supports**. α-Al$_2$O$_3$ hollow fiber supports were prepared by the combined phase-inversion and sintering method in our laboratory[44]. The supports had average outer diameter of 3.6 mm and four cylindrical channels with the same inner diameter of ~0.9 mm. The average pore size and porosity of supports were ~0.14 μm and ~40%, respectively. Before membrane synthesis, the supports were cut into 7 cm or 25 cm pieces and washed thoroughly with DI water. Then the supports were dried at 200 °C overnight.

**DD3R zeolite membranes**. DD3R zeolite membranes were synthesized by the secondary growth method as our previous work[45]. The molar composition was 1-adamantanamine (ADA, 97%, Sigma-Aldrich): $SiO_2$ (Ludox SM-30, Sigma-Aldrich): ethylenediamine (EDA, 99%, Sigma-Aldrich): $H_2O$ = 3: 100: 50: 4000. ADA was first dispersed in an EDA/water mixture at room temperature (25 °C). Next, colloidal silica was dropwise added into the mixture under stirring. The synthesis mixture was aged for 1 h under stirring at room temperature. The hydrothermal crystallization was carried out at 140 °C for 44 h. After that, the as-synthesized membranes were washed and then dried at 200 °C overnight. For the powder preparation, 0.2 wt% ball-milled Sigma-1 zeolite was used as the seed. The as-synthesized DD3R zeolite powders were washed with DI water until the dispersion was neutral and dried at 200 °C overnight.

**The DDR single crystals**. The DDR single crystals were synthesized in our laboratory. Firstly, N-methyltropinium iodide was prepared by adding 25 g methyl iodide (99 wt%, Sigma-Aldrich) dropwise to a solution of 25 g tropine (98 wt%, Alfa Aesar) in 100 g ethanol at 0 °C under stirring and keeping the suspension under reflux for 72 h. The crystalline powder was filtrated and washed with cold ethanol, then dried at 80 °C. For anion-exchange 25 g of methyltropinium iodide was dissolved in 100 mL distilled water and passed ten times through a column filled with 50 g ion-exchange resin. The concentration of N-methyltropinium hydroxide was 0.48 M, which was determined by titration with HCl. The molar ratio of the synthesis mixture was 70 $SiO_2$ (Ludox HS-40, Sigma-Aldrich): 0.2 $Al_2O_3$ (sodium aluminate: 50–56% $Al_2O_3$, Sigma-Aldrich): 17.5 N-methyltropinium hydroxide: 3050 $H_2O$. The precursor was stirred for 30 min before filling into the steel autoclaves with Teflon insets. The crystallisation took place at 160 °C for 42 d.

**Template modulated crystal transition (TMCT)**. The strategy was used for detemplatation. Samples were rapidly heated to 600 °C, 700 °C or 800 °C within 1 min and held for a certain time (1 to 240 min) in iodine tungsten lamp-based furnace (the default condition is heating to 700 °C and then held for 1 min) and cooling down to room temperature within 1 min.

**Conventional calcination (CC)**. CC was conducted for comparison and removal of residual template of TMCT samples. The holding time at 550 °C was 10 h (CC) and at 700 °C was 5 h (CC700) with heating and cooling rates of 0.5 °C min$^{-1}$.

**Gas separation**. Gas separation was carried out in homemade gas separation equipment. For separation of binary equimolar $CO_2/CH_4$, and $SF_6$ single gas permeance, a flow was fed at a rate of about 100 mL·min$^{-1}$, and a helium stream (100 mL min$^{-1}$) was used to sweep the permeate side. A gas chromatography (GC, 7820 A, Agilent Technologies), equipped with thermal conductivity detector (TCD) and a packed column of HAYESEP-DB, was used to analyse gas compositions. The

gas permeance of component i ($_{Pi}$) is defined as:

$$P_i = \frac{F_i}{\Delta p_i} \tag{1}$$

where $F_i$ is the flux of component $i$ (mol m$^{-2}$ s$^{-1}$), and $\Delta_{pi}$ is the transmembrane pressure drop for component $i$ (Pa). The separation selectivity for components i over j is defined as:

$$\alpha = \frac{P_i}{P_j} \tag{2}$$

where $P_i$ and $P_j$ are the gas permeances (mol m$^{-2}$ s$^{-1}$ Pa$^{-1}$) of components, i and j, respectively.

High-pressure $CO_2/CH_4$ separation performance was carried out in a dead-end mode and the gas flow rate was measured by the bubble flow meter. The permeate pressure was kept atmospheric pressure without sweep gas. The gas compositions in permeate and retentate sides were analyzed by a gas chromatograph (GC, 7820 A, Agilent Technologies, with HAYESEP-DB). Considering the variation of gas composition along the membrane (especially at high pressure), we used the following equation to calculate $\Delta P_i$[46].

$$\Delta p_i = \frac{(p_i^{feed} - p_i^{perm}) - (p_i^{ret} - p_i^{perm})}{\ln\left(\frac{p_i^{feed} - p_i^{perm}}{p_i^{ret} - p_i^{perm}}\right)} \tag{3}$$

In light of the kinetic diameter of $SF_6$ (0.513 nm), it is assumed that $SF_6$ diffuses through non-zeolitic pores by Knudsen diffusion and viscous flow.[46] The viscous flow through the membrane only takes place under an absolute pressure drop and in inverse proportion to the gas dynamic viscosity. Therefore, the contribution of non-zeolitic pores to the permeance for other gases is calculated as:

$$P_{Vis,i} = \frac{\eta_{SF_6}}{\eta_i} P_{Vis,SF_6} \tag{4}$$

$$P_{Kn,i} = \sqrt{\frac{M_{SF_6}}{M_i}} P_{Kn,SF_6} \tag{5}$$

$$P_{defect,i} = P_{Kn,i} + P_{Vis,i} \tag{6}$$

Wherein, $P_{Vis,i}$ and $P_{Kn,i}$ is gas permeance contributed by viscous flow and Knudsen diffusion, mol m$^{-2}$ s$^{-1}$ Pa$^{-1}$; $\eta_i$ is dynamic viscosity, μPa s; $M$ is the molecular weight, g mol$^{-1}$.

**Field emission scanning electron microscopy**. FE-SEM (S-4800, Hitachi) was used to recorded the morphologies of DD3R zeolite membranes. To reduce the charging effect, the samples were sputtered with gold for 40 s before characterization.

**X-ray diffraction**. XRD (MiniFlex 600, Rigaku) with Cu $K_\alpha$ radiation ($\lambda = 0.154$ nm at 40 kV and 15 mA) was used to determine membrane phases. The data were recorded from 5° to 50° with a scan speed of 10 min$^{-1}$.

**Confocal laser scanning microscopy**. CLSM (Zeiss LSM 710, Zeiss) was used to detected defects in the zeolite membrane. The membrane side contacted with fluorescein sodium solution ($C_{20}H_{10}Na_2O_5$, Sigma-Aldrich) while the support side contacted with DI water. The soaking duration is 12 h at room temperature.

**Thermogravimetric analyzer**. TGA (TG209F1, NETZSCH) was used to determine the degree of detemplation, and performed under airflow with a heating rate of 10 °C min$^{-1}$.

We defined the mass loss of fresh sample between 200 and 1000 °C as cage occupation of template in zeolite cages. The cage occupation of different stage sample can be calculated by the following equation:

$$\text{cage occupation}_i = \frac{\Delta m_i}{\Delta m_{fresh}} \times 100\% \tag{7}$$

where $\Delta m_i$ is mass loss of different state sample between 200 and 1000 °C.

**The N$_2$ (77K) adsorption isotherms**. The N$_2$ (77K) adsorption isotherms were recorded in Micromeritics ASAP2460. Prior to adsorption measurements, the samples were outgassed at 300 °C for 8 h under turbomolecular pump vacuum.

**The X-ray photoelectron spectroscopy**. XPS were recorded by Thermo ESCA-LAB 250XI equipped with Al $K_\alpha$ X-ray source at ambient temperature and chamber pressure of about $8 \times 10^{-10}$ Pa. The X-ray gun was operated at 16 mA and 12.5 kV. All the spectra measured were corrected by setting the reference binding energy of carbon (C 1 $s$) at 284.80 eV.

**Solid-state $^{13}$C MAS NMR spectra**. Solid-state 13C MAS NMR spectra were acquired by Agilent 600 M. The sample was packed in a 4 mm zirconia rotor and recorded on a Bruker DSX-300 spectrometer at ambient temperature.

**The thermal expansion**. The thermal expansion of support sample was acquired by DIL803 dual sample dilatometer (TA Instrument). The sample was heating from 50 °C to 950 °C with heating rate of 5 °C min$^{-1}$ at air atmosphere.

**The high-temperature in-situ powder X-ray diffraction data**. The high-temperature in-situ powder X-ray diffraction data was collected on an Empyrean Bragg-Brentano diffractometer (Malvern Panalytical B. V., The Netherlands) with a Cu $K_\alpha$ X-ray tube equipped with iCore and dCore optics. The automated optical module of iCore combined with a Soller 0.03 radians was used to provide an incident beam path, which had 450 eV energy resolution. A programmable divergence slit was adopted to optimize the beam divergence. A fixed slit mode with the optimal resolution was utilized for the measurement. The other automated optical module, dCore, provides a path for the diffracted beam, wherein the Soller slit was 0.02 radians. The X-ray was detected with a position-sensitive PIXcel$^{3D}$. A HTK1200N chamber provided by AntonPaar was used to heat the zeolite powder from 25 °C to 900 °C in air. After 5 min equilibrium at each temperature, the X-ray diffraction was collected for 17 min. A Pawley fit was used to calculate the cell parameter of zeolite crystals. The height of sample position was corrected to compensate the effect of flange expansion during the heating up. This was accomplished by measuring the direct beam and translating the sample until the beam was completely blocked. Afterwards, the inflection point was determined automatically and the height was adjusted to the right position for each temperature. The correction above ensures the independence of temperature on the sample displacement, which is critical for the cell parameters determination with high accuracy. The HighScore suite was used to refine the curves in batch and multi-cycle mode[47]. Finally, the refined cell parameters and error bars were exported in ascii format.

**Single-crystal X-ray diffraction**. Single-crystal X-ray diffraction experiments on fresh, TMCT and TMCT + CC DDR zeolites were performed at the XRD1 beamline of the Elettra Synchrotron facility (CNR Trieste, Basovizza, Trieste, Italy)[48]. Suitable crystals were isolated and mounted on a MiTeGen loop in a droplet of NVH oil. Diffraction data were acquired on a Pilatus 2 M detector using a monochromatic 0.6199 Å wavelength at −173 °C. Low-temperature conditions were obtained by using a cold dry nitrogen stream produced with an Oxford Cryostream 700 [Oxford Cryosystems Ltd., Oxford, United Kingdom). Frames were collected running a phi scan with oscillation 0.5° and covering an angular range of 360°. The data have been processed using the CrysalisPro software package version 1.2.10.44. [Rigaku Oxford Diffraction, CrysAlisPro Software System, Version 1.171.38.43, Oxford, UK, 2017]. Structure solution and refinement were performed by the programs ShelXT[49] and ShelXL[50] (least squares) respectively, using the Olex2 software[51] version 1.2.10. The crystal structure files for the described DDR crystals in the fresh, TMCT and TMCT + CC forms have been deposited in the Cambridge Crystallographic Data Centre with deposition number 1979764, 1979765, and 1979766, respectively.

## Data availability

Data generated in this study are provided in the manuscript or supplementary information and source data are provided with this paper. The crystal structure data are available in the Cambridge Crystallographic Data Centre under deposition number 1979764, 1979765, and 1979766, respectively. Source data are provided with this paper.

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

## Acknowledgements
This work has been supported by the National Natural Science Foundation of China (22035002, 21908097, 22008111), National Key R&D Program of China (2021YFC2101200), State Key Laboratory of Materials Oriented Chemical Engineering (ZK202002), Jiangsu Specially-Appointed Professors Program and Postgraduate Research & Practice Innovation Program of Jiangsu Province (KYCX21_1168). The Elettra Synchrotron facility (CNR Trieste, Basovizza, Italy) is acknowledged for granting beamtime at the single-crystal diffraction beamline XRD1 (20185483) and the beamline staff is gratefully thanked for the assistance. We thank W. Yang from Delft University of Technology for the synthesis of *N*-methyltropinium hydroxide and F. H. from Tsinghua University for the Confocal Laser Scanning Microscopy characterization. S.C. gratefully thanks Prof. Monique A. van der Veen for supporting his contribution to this study.

## Author contributions
X.G. proposed the research and supervised the project. P.D. performed membrane fabrication, SEM, BET, NMR, PXRD, TGA characterization and gas separation performance evaluation. Y.Z. conceived the idea and contributed fruitful discussion. X.W. synthesized the single crystal, coordinated the structural refinement. S.C. conducted single-crystal X-ray diffraction experiments and solved the crystal structure. G.N. performed in-situ various temperature PXRD and solved the unit cell parameters. Z.H. contributed the hollow fiber supports. P.D., X.W., Y. Z., W.J., and X.G. finalized and revised the manuscript. All authors discussed the results and commented on the manuscript.

## Competing interests
The authors declare no competing interests.
