## [Peer Review File · Nature Communications]

REVIEWER COMMENTS

Reviewer #1 (Remarks to the Author):

This work demonstrated the use of TMCT method to prepare zeolite membrane for effective CO₂/CH₄ separation and obtained outstanding separation efficiency. The authors used a series of characterizations to unveil the properties of the as-prepared materials and the TMCT mechanism. Indeed, this work is of great potential importance as it could advance the industrial application of zeolite membranes for gas separation. To make this work sufficiently novel and important to reach the standard of Nature Comm, the overarching question the authors need to address is can this exciting method be generally applied to other zeolites to make high-performance membranes. Apart from it, some other questions for consideration are as follows. I would recommend to accept this manuscript if all my questions can be well addressed.

1. Many zeolites are known to undergo negative thermal expansion when templates are absent. Can the authors discuss in detail the thermal expansion behaviour of DD3R by comparing it with literature, if applicable, and the effect of the template's presence and removal on the thermal expansion behaviour?
2. When the authors discussed the good performance of the DD3R membrane for CO₂/CH₄ separation, please also compare with other reported inorganic membranes made of zeolites, MOFs, and carbons in terms of both selectivity and permeability.
3. According to the author, the condition for TMCT is "Remarkably the membranes kept intact by an instantaneous overheating at 700 °C for one minute (TMCT)". Then the authors used the in situ XRD to unveil the TMCT mechanism. However, the condition used in XRD experiments was not consistent with the real calcination process, i.e., "DD3R zeolite powders were subject to heating from room temperature to 900 °C step by step with 5 minutes equilibrium time prior to the start of the 17 minutes data collection for cell parameters determination". Different calcination duration or ramping rate can affect a lot the properties of the samples. Do the authors think the XRD data is comprehensive enough to explain the mechanism? In addition, what is the atmosphere during TMCT? Is it consistent with that of in situ XRD?
4. How is the desorption kinetics?
5. What is the reusability of this material?

Reviewer #2 (Remarks to the Author):

This work reports significant progress in preparing zeolite membranes with reduced defects. The approach is sound and novel and is supported by X-ray diffraction and NMR data. I believe it can be published in Nature Communications after the following minor revisions:

1. The authors should prominently provide the range of selectivities in the abstract and text. Currently, they provide the best membrane performance (see also comment 7).
2. The authors may want to cite the paper that introduced confocal microscopy as a method to assess defects in zeolite membranes: Fluorescence confocal optical microscopy imaging of the grain boundary structure of zeolite MFI membranes made by secondary (seeded) growth

Bonilla, G; (...); Xomeritakis, G

Feb 15 2001 | JOURNAL OF MEMBRANE SCIENCE 182 (1-2) , pp.103-109

3. I think that the obtained flux of 0.01 mol/m²-s at 31 bara is still on the lower side for commercial application. The authors may want to comment on membrane area required for a typical plant, the volume and number of the modules, and associated capital cost.

4. I believe the caption of Figure 5 d is wrong. Please check.

5. What is the composition of the binary mixture used as feed for the data of Figure 5d (which includes 31 bara feed experiments)? Was it equimolar?

6. It would be important addition if the authors could provide permeation data of mixtures at high feed pressure (31 bara) and undiluted permeate for a range of feed compositions. This will provide useful data for assessing potential for practical use (across the range that a membrane will encounter in a process).

7. The performance of the membrane (flux and separation factor) under field conditions (31 bara) should be highlighted in the abstract and text. Currently, only the highest performance is mentioned and this could be misleading if one does not pay attention to the fine print.

8. The term defect-free should be avoided. By their own assessment the authors' membranes have defects. Maybe the term low-defect-density or something similar is more appropriate.

Point-by-point Response to Reviewers' Comments

We thank the reviewers for their critical evaluation and constructive suggestions. Below we address the itemized comments and provide our response with the updated text, figures, tables and relevant literature references.

The reviewers' comments are given in italics, our response in normal font.

The main text is given in blue font, while changes are indicated by italics and blue font.

Reviewer 1:

This work demonstrated the use of TMCT method to prepare zeolite membrane for effective CO₂/CH₄ separation and obtained outstanding separation efficiency. The authors used a series of characterizations to unveil the properties of the as-prepared materials and the TMCT mechanism. Indeed, this work is of great potential importance as it could advance the industrial application of zeolite membranes for gas separation.

To make this work sufficiently novel and important to reach the standard of Nature Comm, the overarching question the authors need to address is can this exciting method to be generally applied to other zeolites to make high-performance membranes. Apart from it, some other questions for consideration are as follows. I would recommend to accept this manuscript if all my questions can be well addressed.

Response

Thanks for reviewer's positive comments. All the questions raised from the reviewer were well addressed in the following response and partial response was incorporated to the revised manuscript. We believe the significant revision satisfies the reviewer.

Regarding the feasibility to other zeolite membranes, we are working to collect more detailed information. Here we would like to share some preliminary results of MFI and SSZ-13 zeolite membranes (Table R1). **(1) MFI zeolite membranes:** Two MFI zeolite membranes were detemplated with conventional calcination (CC) and TMCT method, respectively. In the case of CC-treated membranes (M1 and M2), the selectivity of *p*-xylene (PX) over *o*-xylene (OX) were 11 and 20. However, the

selectivity was up to 45 and 82 if TMCT method was adopted (M3 and M4). This would be another evidence to our proposal that the diffusion resistance of decomposed species is one of key points for detemplation. MFI zeolite has larger effective pore size (0.55×0.51 nm and 0.56×0.53 nm) than DD3R zeolite (0.36×0.44 nm). Its template (tetrapropylammonium cations, TPA⁺) can be easily removed by sole TMCT treatment at 700 °C for 1 minute. In that case, the PX permeance of TMCT-treated membrane was comparable to that of the CC-treated one. **(2) SSZ-13 zeolite membranes:** Similar experiment was conducted on SSZ-13 zeolite membranes (M7 and M8, Table R1). After TMCT treatment, the membranes showed CO₂/CH₄ selectivity of 11 and 22, while the CO₂ permeance was 3.4×10^{-8} mol m⁻² s⁻¹ Pa⁻¹ and 5.1×10^{-8} mol m⁻² s⁻¹ Pa⁻¹. Interestingly, the CO₂ permeance increased 5-fold and reached to 2.4×10^{-7} mol m⁻² s⁻¹ Pa⁻¹ after the following CC treatment. Simultaneously, the CO₂/CH₄ selectivity increased up to 100 even higher than the conventional calcinated membranes (M5 and M6). The results matched well with DD3R zeolite membranes as demonstrated in the Revised Manuscript. Therefore, the TMCT approach was universally efficient to other zeolite membranes (e.g., MFI and SSZ-13).

Table R1 Separation performance of MFI and SSZ-13 zeolite membranes detemplated by convention calcination (CC) and TMCT method.

No	Membrane	Detemplation	Separation performance		
			Feed	$P_1 / \text{mol m}^{-2} \text{s}^{-1} \text{Pa}^{-1}$	α
M1	MFI	CC450 ^c	PX/OX ^a	2.6×10^{-8}	11
M2	MFI	CC450 ^c	PX/OX ^a	2.1×10^{-8}	20
M3	MFI	TMCT ^d	PX/OX ^a	2.6×10^{-8}	82
M4	MFI	TMCT ^d	PX/OX ^a	4.7×10^{-8}	45
M5	SSZ-13	CC450 ^c	CO ₂ /CH ₄ ^b	2.4×10^{-7}	77
M6	SSZ-13	CC450 ^c	CO ₂ /CH ₄ ^b	2.6×10^{-7}	81
M7	SSZ-13	TMCT ^d	CO ₂ /CH ₄ ^b	3.4×10^{-8}	11
		TMCT ^d +CC450 ^c	CO ₂ /CH ₄ ^b	2.0×10^{-7}	41
M8	SSZ-13	TMCT ^d	CO ₂ /CH ₄ ^b	5.1×10^{-8}	22
		TMCT ^d +CC450 ^c	CO ₂ /CH ₄ ^b	2.4×10^{-7}	100

^a: both components had partial pressure of 2 kPa at 150 °C; ^b: equimolar CO₂/CH₄ binary mixture at 25 °C and feed pressure of 2 bara; ^c: calcination at 450 °C for 12 h with heating and cooling rate of 0.5 °C min⁻¹; ^d: TMCT at 700 °C for 1 min.

Comment 1

Many zeolites are known to undergo negative thermal expansion when templates are absent. Can the authors discuss in detail the thermal expansion behaviour of DD3R by comparing it with literature, if applicable, and the effect of the template's presence and removal on the thermal expansion behaviour?

Response

Thanks for the reviewer's good suggestion. Actually, DD3R zeolite undergoes shrinkage if its template is removed, which is confirmed by the decreased unit cell volume from 6800 Å³ to 6776 Å³ in our manuscript. The negative thermal expansion behavior of template-free DD3R zeolite was well investigated in literature (Table R2)^{38, 39, 40}. The table was incorporated to the revised ESI as Supplementary Table S2. Herein we are emphasizing the thermal expansion behavior of template occupied DD3R zeolite, which could well explain the crack formation during conventional calcination. This was well addressed in the revised manuscript: "To quantify the thermal expansion, the unit cell volume was determined and maximized at 425 °C (Fig. 4b). However, the temperature required to reach the maximum unit cell volume decreased to 325 °C for TMCT sample and 275 °C for empty sample (Fig. 4b and Supplementary Fig. S22~Fig. S24). Most importantly, both TMCT and empty samples showed less thermal expansion, only 45.9% of the fresh one (see thermal expansion coefficients from Supplementary Table S2)."

Table R2 Thermal expansion coefficients of different zeolites.

Zeolite	Stage	Temp. / °C	Thermal expansion coefficient / K ⁻¹	Ref
DD3R	Fresh zeolite	25-425	1.87×10 ⁻⁵	This work
	TMCT zeolite	25-325	1.14×10 ⁻⁵	
	Empty zeolite	25-275	1.46×10 ⁻⁵	
		275-895	-3.87×10 ⁻⁶	
DD3R	Empty zeolite	<300	1.8×10 ⁻⁵	38
	Empty single crystal	<227	1.4×10 ⁻⁵	
	Empty zeolite	>300	-3×10 ⁻⁶	
DD3R	Empty zeolite	25-219	3.51×10 ⁻⁵	39
	Empty zeolite	219-912	-8.7×10 ⁻⁶	

MFI	Empty zeolite	25-75	2.7×10^{-5}	39
		120-702	-1.51×10^{-5}	
CHA	Empty powder	20-600	-2.85×10^{-5}	40

The supporting information was revised as follow: “The thermal expansion coefficient was defined by the following equation¹:

$$\beta_v = \frac{\Delta V}{V_0 \times \Delta T} \quad (1)$$

where β_v represents the volume thermal expansion coefficient, ΔV is the change in crystal volume, V_0 is the initial crystal volume, and ΔT is the temperature difference.”

Reference

1. Liu Z, Gao Q, Chen J, Deng J, Lin K, Xing X. Negative thermal expansion in molecular materials. Chem Commun 54, 5164-5176 (2018).
- 38 Kajihara K, et al. Twinning by Merohedry and Thermal Expansion of Zeolitic Clathrasil Deca-dodecasil 3R. Inorg Chem 59, 5600-5609 (2020).
- 39 Park SH, Kunstleve RWG, Graetsch H, Gies H. The thermal expansion of the zeolites MFI, AFI, DOH, DDR, and MTN in their calcined and as synthesized forms. In: Progress in Zeolite and Microporous Materials, Pts a-C (eds Chon H, Ihm SK, Uh YS) (1997).
- 40 Lightfoot P, Woodcock DA, Maple MJ, Villaescusa LA, Wright PA. The widespread occurrence of negative thermal expansion in zeolites. J Mater Chem 11, 212-216 (2001).

Comment 2

When the authors discussed the good performance of the DD3R membrane for CO₂/CH₄ separation, please also compare with other reported inorganic membranes made of zeolites, MOFs, and carbons in terms of both selectivity and permeability.

Response

Thanks. The CO₂/CH₄ separation performance of other reported inorganic

membranes made of zeolites, MOFs, and carbons were plotted in the revised Fig. 5c. The revised figure was shown below as Fig. R1 for a quick view. The main text was revised as well: “The CO₂/CH₄ separation performance of DD3R zeolite membranes obtained in this work far surpassed the well-known “upper bound” defined by polymeric membranes²⁸ (e.g., polyimide²⁹, Matrimid³⁰ and the commercial cellulose acetate³¹). Meanwhile the selectivity was beyond the calcinated zeolite membranes (SAPO-34³², SSZ-13³³ and AIPO-18³⁴), carbon molecular sieve (CMS) membranes³⁵ and ZIF-8 membranes³⁶ (Fig. 5c).”

Fig. R1 Comparison on CO₂/CH₄ separation performance, involving commercial cellulose acetate (CA) membranes³¹, polyimide membranes²⁹, Matrimid membranes³⁰, carbon molecular sieve (CMS) membranes³⁵, ZIF-8 membranes³⁶ and other 8MR zeolite membranes detemplated by conventional calcination^{28, 32, 33, 34}. (Note: Figure R1 was shown as Fig. 5c in the revised manuscript.)

Reference

- 32 Li S, Falconer JL, Noble RD. Improved SAPO-34 membranes for CO₂/CH₄ separations. *Adv Mater* **18**, 2601-2603 (2006).
- 34. Wu T, Wang B, Lu Z, Zhou R, Chen X. Alumina-supported AIPO-18 membranes for CO₂/CH₄ separation. *J Membr Sci* **471**, 338-346 (2014).
- 35. Lei L, Lindbråthen A, Hillestad M, Sandru m, Favvas EP, He X. Screening cellulose spinning parameters for fabrication of novel carbon hollow fiber

membranes for gas separation. *Ind Eng Chem Res* 58, 13330-13339 (2019).

36. Babu DJ, et al. Restricting lattice flexibility in polycrystalline metal-organic framework membranes for carbon capture. *Adv Mater* 28, 1900855 (2019).

Comment 3

According to the author, the condition for TMCT is “Remarkably the membranes kept intact by an instantaneous overheating at 700 °C for one minute (TMCT)”. Then the authors used the in situ XRD to unveil the TMCT mechanism. However, the condition used in XRD experiments was not consistent with the real calcination process, i.e., “DD3R zeolite powders were subject to heating from room temperature to 900 °C step by step with 5 minutes equilibrium time prior to the start of the 17 minutes data collection for cell parameters determination”. Different calcination duration or ramping rate can affect a lot the properties of the samples. Do the authors think the XRD data is comprehensive enough to explain the mechanism? In addition, what is the atmosphere during TMCT? Is it consistent with that of in situ XRD?

Response

Thanks for the good suggestion. Indeed the XRD data will be more comprehensive if we can collect the diffraction signal under the identical condition as TMCT. Limited to the facility in our group and cooperator’s lab, we have to spend 17 minutes to collect the X-ray diffraction signal for crystal structure refinement. I would say the XRD data is comprehensive enough before the temperature of 510 °C, wherein the templates begin to decompose (Supplementary Figure S8). However, the zeolite property would be affected more by the calcination duration and ramping rate at temperature higher than 510 °C because of the template starts to decompose. That is the reason why different samples (fresh, TMCT-treated and detemplated DD3R zeolite) was used for structure refinement in the manuscript. All the *in-situ* XRD atmosphere was the same as that of TMCT. To make it clear, the Method Section was revised as follow: “**Template modulated crystal transition (TMCT) strategy was used for detemplation in air atmosphere**” and “**The temperature dependence of the materials was investigated using a HTK1200N chamber from AntonPaar in the range**

of 25-1000 °C *in air atmosphere*.”

Comment 4

How is the desorption kinetics?

Response

The desorption kinetics involve two circumstance: The decomposition products of templates from zeolite cages and the permeate gas from membrane surface. Generally, the template molecules decomposed to NH₃, CO₂, C₂H₄, and C₃H₆ in the inert atmosphere^{1, 2}. Herein, Temperature Program Desorption coupled with Mass Spectroscopy (TPD-MS) was used to determine the decomposition products of templates trapped in DD3R zeolite. The signal at $m/e=2$, 16, 17, 28 and 42 was assigned to H₂, CH₄, NH₃, C₂H₄ and C₃H₆^{1, 2}, respectively. The signal intensity was plotted with time as Fig. R2. The template decomposition started from ~504 °C and maximized at ~642 °C, which is consistent with TGA results in the main text. The application to natural gas upgrading mainly contains impurities of CO₂, CH₄, also C₂H₆ and C₃H₆^{3, 4}. Therefore, the desorption kinetics of H₂, CO₂, CH₄, C₂H₄, C₂H₆ and C₃H₆ would be the concerns from reviewer.

Zero length column (ZLC) method was used to determine the desorption kinetics of adsorbate from pre-saturated zeolite⁵. As shown in Fig. R3, the gas uptake gradually decreased with sweeping time. Based on these curves, the gas diffusivity in zeolitic pores could be extracted and summarized in Table R3. In other words, the diffusivity would be another parameter to describe the desorption kinetic. In this sense, the desorption of bulky molecules would be much slower (*e.g.*, C₂H₄, C₂H₆ and C₃H₆) than that of smaller ones (*e.g.*, H₂, CO₂ and CH₄). The desorption can be speed up by elevating the temperature.

Fig. R2 TPD-MS spectrum of fresh DD3R zeolite with heating rate of 50 °C min⁻¹.

Fig. R3 ZLC response curves for methane, ethane and ethylene.

Table R3 Diffusivities of hydrocarbon through DDR zeolite determined by ZLC method.

$T / ^\circ\text{C}$	$D_i / \times 10^{-12} \text{ m}^2 \text{ s}^{-1}$			
	CH_4^6 (0.38 nm)	C_2H_4^5 (0.42 nm)	C_2H_6^5 (0.44 nm)	C_3H_6^7 (0.47 nm)
0	0.29	-	0.06	-
25	0.8	0.10	0.11	-
50	1.64	0.15	0.14	0.05
75	3.4	0.23	0.23	0.12
100	-	0.29	0.27	0.21
150	-	-	-	0.54

Reference

- Gao X, Yeh CY, Angevine P. Mechanistic study of organic template removal from ZSM-5 precursors. *Microporous Mesoporous Mater* **70**, 27-35 (2004).
- Zhang Y, Tokay B, Funke HH, Falconer JL, Noble RD. Template removal from SAPO-34 crystals and membranes. *J Membr Sci* **363**, 29-35 (2010).

- den Exter MJ, Jansen JC, van Bekkum H. Separation of permanent gases on the all-silica 8-ring clathrasil DD3R. In: *Studies in Surface Science and Catalysis* (ed[^](eds Weitkamp J, Karge HG, Pfeifer H, Hölderich W). Elsevier (1994).
- Zhu W, Kapteijn F, Moulijn JA, Jansen JC. Selective adsorption of unsaturated linear C₄ molecules on the all-silica DD3R. *Phys Chem Chem Phys* **2**, 1773-1779 (2000).
- Vidoni A, Ruthven DM. Diffusion of C₂H₆ and C₂H₄ in DDR zeolite. *Ind Eng Chem Res* **51**, 1383-1390 (2012).
- Vidoni A, Ruthven D. Diffusion of methane in DD3R zeolite. *Microporous Mesoporous Mater* **159**, 57-65 (2012).
- Vidoni A. Adsorption and diffusion of light hydrocarbons in DDR zeolite. (University of Maine, 2011).

Comment 5

What is the reusability of this material?

Response

Thanks. DD3R zeolite membranes are stable for long-term operation. Previously we demonstrated the hydrothermal stability of DD3R zeolite membranes in humid atmosphere for >300 h¹ and in 100 ppm H₂S atmosphere for >1000 h². Up to now, the membrane used for high-pressure gas separation test in Fig. 5d was stored in our lab for two years. To demonstrate the reusability, the membrane performance was evaluated again. Strikingly the membrane exhibited CO₂ permeance of 3.08×10^{-8} mol m⁻² s⁻¹ Pa⁻¹ and CO₂/CH₄ selectivity of 44 for equimolar CO₂/CH₄ separation at 31 bara, which is almost the same as the initial performance (Supplementary Fig. S32). Thus, we believe DD3R zeolite membranes can be well reused even after the long-term storage.

Table R4 CO₂/CH₄ separation performance of DD3R zeolite membrane before and after 2-year storage in air atmosphere. (Table R4 was shown as Supplementary Table S6 in the revised manuscript)

Membrane	Num.	Separation performance			
		Feed	Pressure / bara	$P_i / \text{mol m}^{-2} \text{s}^{-1} \text{Pa}^{-1}$	α
As-synthesized	1	CO ₂ /CH ₄	31 bara	3×10^{-8}	43
	2	CO ₂ /CH ₄	31 bara	4.1×10^{-8}	51
2-year storage	1	CO ₂ /CH ₄	31 bara	3.08×10^{-8}	44
	2	CO ₂ /CH ₄	31 bara	4.2×10^{-8}	49

On the other hand, the performance recovery from membrane fouling should be another criterion to evaluate the reusability. In our recent publication³, DD3R zeolite membranes were used to extract helium from natural gas. Both permeance and selectivity decreased if ethane was added into the feed (Fig. R4), indicating the zeolitic pores were blocked in some extent. However, the permeance completely recovered once ethane was cut off. This is another evidence to prove the reusability of our DD3R zeolite membranes.

Fig. R4 Long-term stability of DD3R zeolite membrane for 2: 98 He/CH₄ mixture separation. The orange and purple symbols represent gas separation performance with and without 3.6 % ethane in the feed. Measured at 298 K and 0.7 MPa. Ar used as the sweep gas.³

Reference

1. Wang X, *et al.* Xenon recovery by DD3R zeolite membranes: Application in Anaesthetics. *Angew Chem Int Ed* **58**, 15518-15525 (2019).
2. Du P, *et al.* Efficient scale-up synthesis and hydrogen separation of hollow fiber DD3R zeolite membranes. *J Membr Sci* **636**, 119546 (2021).
3. Zhang P, *et al.* Helium extraction from natural gas using DD3R zeolite membranes. *Chinese Journal of Chemical Engineering*. Preprint at <https://doi.org/10.1016/j.cjche.2021.09.004>, (2021).

Reviewer 2:

This work reports significant progress in preparing zeolite membranes with reduced defects. The approach is sound and novel and is supported by X-ray diffraction and NMR data. I believe it can be published in Nature Communications after the following minor revisions.

Response

Thanks for the positive comments. All the questions raised from the reviewer were well addressed in the following response and partial response was incorporated in the updated manuscript. We believe the significant revision will satisfy the reviewer.

Comment 1

The authors should prominently provide the range of selectivities in the abstract and text. Currently, they provide the best membrane performance (see comment 7).

Response

Thanks. As suggested by the reviewer, the abstract was revised as follow: “The resulting membranes exhibited CO_2/CH_4 selectivity of 157 ~ 1,172 and CO_2 permeance of $(890 \sim 1,540) \times 10^{-10} \text{ mol m}^{-2} \text{ s}^{-1} \text{ Pa}^{-1}$. The performance far surpassed the state-of-the-art membranes for nature gas upgrading. $The CO_2$ flux and CO_2/CH_4 mixture selectivity reached to $3.6 \text{ Nm}^3 \text{ m}^{-2} \text{ h}^{-1}$ and 43 even at feed pressure up to 31 bara. Such strategy paves the way of all-silica zeolite membranes to practical applications.” We also emphasized the range of selectivity in the main text as follow: “The CO_2/CH_4 selectivity ranged from 157 to 1,172, giving an averaged value of 422”.

Comment 2

The authors may want to cite the paper that introduced confocal microscopy as a method to assess defects in zeolite membranes: Fluorescence confocal optical microscopy imaging of the grain boundary structure of zeolite MFI membranes made by secondary (seeded) growth Bonilla, G; (...); Xomeritakis, G Feb 15 2001 / JOURNAL OF MEMBRANE SCIENCE 182 (1-2), pp.103-109

Response

Thanks. The literature has been cited and incorporated into the main text as follow: “The non-selective macro-pores were visualized by confocal laser scanning microscopy (also called fluorescence confocal optical microscopy¹⁸) using fluorescein sodium (~ 1 nm) as the probe molecules (Supplementary Fig. S4), which exclusively penetrate through and retain in the non-zeolitic macro-pores.”

Reference

18 Bonilla G, Tsapatsis M, Vlachos DG, Xomeritakis G. Fluorescence confocal optical microscopy imaging of the grain boundary structure of zeolite MFI membranes made by secondary (seeded) growth. J Membr Sci 182, 103-109 (2001).

Comment 3

I think that the obtained flux of 0.01 mol m⁻² s⁻¹ at 31 bara is still on the lower side for commercial application. The authors may want to comment on membrane area required for a typical plant, the volume and number of the modules, and associated capital cost.

Response

Actually, the CO₂ flux was 0.1 mol m⁻² s⁻¹ at 31 bara rather than 0.01 mol m⁻² s⁻¹ as claimed by the reviewer. The separation performance of our membranes and the commercial polymeric membranes was commented here to show the potential application. As stated by Baker *et al.*¹, the CO₂/CH₄ selectivity and CO₂ permeance of the current commercial membranes are 12 and 100 GPU. The membrane with doubled selectivity and permeance (24 and 200 GPU) would become next-generation membranes¹. Our membrane exhibited selectivity of >157 and permeance of >270

GPU far surpassing the criterion as mentioned above. Therefore, our membranes are potential to practical application in the view of membrane performance.

Ref 2 reported a typical natural upgrading plant using polymeric membranes. The capacity was 50 MMscfd raw natural gas containing 30% CO₂. Because of the undesirable selectivity, a two-stage process was required to endure a reliable methane recovery (Fig. R5a). However, one-stage process was fine in the case of our high-selective membranes (Fig. R5b). Meanwhile the operation pressure can be up to 55 bara since our membranes were featured with antiplasticization. Therefore, the required membrane area is only half of the commercial polymer membrane.

The manuscript was revised as follow: *“It can be anticipated that the hollow fiber DD3R zeolite membranes can satisfy the requirement of natural gas upgrading for 30% CO₂ in one-stage separation with CH₄ loss less than 3%, which would reduce the desired membrane area and separation cost^{3, 4}.”*

Fig. R5 (a) Flow design of a combination two-stage and one-stage membrane system using commercial polymer membrane to remove CO₂ from 30% CO₂ concentration

gas on an offshore platform². **(b)** Flow scheme of one-stage DD3R zeolite membrane separation plants to remove 30% CO₂ from natural gas.

Reference

- Galizia M, Chi WS, Smith ZP, Merkel TC, Baker RW, Freeman BD. 50th anniversary perspective: Polymers and mixed matrix membranes for gas and vapor separation: A review and prospective opportunities. *Macromolecules* **50**, 7809-7843 (2017).
- Baker RW, Lokhandwala K. Natural gas processing with membranes: An overview. *Ind Eng Chem Res* **47**, 2109-2121 (2008).

Comment 4

I believe the caption of Figure 5 d is wrong. Please check.

Response

Thanks. Fig. 5d was shown as Fig. R6 here for a quick view. The caption was revised as follow: “Gas permeation flux (F) and CO₂/CH₄ selectivity of 20 cm hollow fiber DD3R zeolite membrane in pure components (open symbols) and equimolar mixture (solid symbols) with pressure range of 3 to 31 bara (condition: room temperature, without sweep gas).”

Fig. R6 Gas permeation flux (F) and CO₂/CH₄ selectivity of 20 cm hollow fiber DD3R zeolite membrane in pure components (open symbols) and equimolar mixture (solid symbols) with pressure range of 3 to 31 bara (condition: room temperature,

without sweep gas).

“The gas permeance of component i (P_i) is defined as:

$$P_i = \frac{F_i}{\Delta p_i} \quad (1)$$

where F_i is the flux of component i ($\text{mol m}^{-2} \text{s}^{-1}$), and Δp_i is the transmembrane pressure drop for component i (Pa).”

Comment 5

What is the composition of the binary mixture used as feed for the data of Figure 5d (which includes 31 bara feed experiments)? Was it equimolar?

Response

Thanks. Fig. 5d was shown as Fig. R6 in last response for a quick view. The caption has been revised to well describe the feed composition. Meanwhile the detailed composition was given in the Method Section and supporting information as follow: “**Gas separation** was carried out in home-made gas separation equipment. For separation of equimolar CO₂/CH₄.” “Supplementary Fig. S30 Equimolar CO₂ and CH₄ binary gas separation performance as a function of feed pressure of DD3R zeolite membrane.” “Supplementary Table S4 Contribution of surface diffusion (zeolitic cages), Knudsen diffusion and viscous flow of CO₂ and CH₄ permeation through DD3R zeolite membranes in Fig. 5d.”

Comment 6

It would be important addition if the authors could provide permeation data of mixtures at high feed pressure (31 bara) and undiluted permeate for a range of feed compositions. This will provide useful data for assessing potential for practical use (across the range that a membrane will encounter in a process).

Response

Thanks. As suggested, we tested the CO₂/CH₄ mixture separation at 31 bara for a range of feed compositions. The results are indeed useful for the practical process design since the permeate was not diluted. The results were shown as Supplementary Fig. S32 in the revised ESI and Fig. R7 below for a quick view. The manuscript was

also revised as follow: “Thus, CH₄ permeance through zeolite cages increased with the feed pressure. Meanwhile CO₂/CH₄ selectivity with different feed compositions ranged from 39 to 50 at feed pressure of 31 bara (Supplementary Fig. S32).”

Fig. R7 Effect of molar fraction on CO₂/CH₄ mixture separation at 31 bara. No sweep gas. [Note: shown as Supplementary Fig. S32 in the revised ESI.]

Comment 7

The performance of the membrane (flux and separation factor) under field conditions (31 bara) should be highlighted in the abstract and text. Currently, only the highest performance is mentioned and this could be misleading if one does not pay attention to the fine print.

Response

Thanks. The high-pressure performance was emphasized in the abstract as follow: “The CO₂ flux and CO₂/CH₄ mixture selectivity reached to 3.6 Nm³ m⁻² h⁻¹ and 43 even at feed pressure up to 31 bara.” Meanwhile the main text was also revised to emphasize the high pressure: “When the feed pressure was up to 31 bara, CO₂ flux was 8.1 Nm³ m⁻² h⁻¹ for single-gas permeation and 3.6 Nm³ m⁻² h⁻¹ for binary mixture with a mixture selectivity of 43.”

Comment 8

The term defect-free should be avoided. By their own assessment the authors' membranes have defects. Maybe the term low-defect-density or something similar is more appropriate.

Response

Thanks. The abstract and main text have been revised as follow:

(1) “The resulting membranes exhibited CO₂/CH₄ selectivity of 157 ~ 1,172 and CO₂ permeances of (890~1,540) × 10⁻¹⁰ mol m⁻² s⁻¹ Pa⁻¹.”

(2) “All the fresh DD3R membranes showed CO₂ permeance less than 0.1 × 10⁻¹⁰ mol m⁻² s⁻¹ Pa⁻¹ for binary CO₂/CH₄ mixture since they were to be of high quality and their cavities were occupied with templates (Fig. 5a).”

(3) “Zeolitic pores contributed 99.99% CO₂ permeance even under 31 bara (Supplementary Fig. S30), confirming the membranes were to be of high quality.”

REVIEWER COMMENTS

Reviewer #1 (Remarks to the Author):

Generally, the authors have addressed my comments in the last round. Some further suggestions are as follows:

- 1) In my view, the promising preliminary results of MFI and SSZ-13 zeolite membranes should be presented in the supporting info of this submission, because it is vitally important to demonstrate the general applicability of this new method.
- 2) As for comparing with other reported inorganic membranes made of zeolites, MOFs, and carbons in terms of both selectivity and permeability, the authors are suggested to list best-performers in the literature for a fair comparison.
- 3) As the authors acknowledge the limitation of XRD, they are advised to revisit their elucidation of the mechanism accordingly.

The manuscript can be accepted if above comments can be addressed.

Reviewer #2 (Remarks to the Author):

I do not agree with the reply of the authors on my comment regarding flux. The 0.1 mol/m²-s flux that the authors mention is for pure CO₂ feed at 31 bar with small stage cut. In the actual process, even if we consider 55 bar with 30% CO₂ feed and 1 bar permeate (the second case in the reply of the authors), one has to account for the retentate being only 2% in CO₂. At this large stage cut, the membrane will operate eventually at low driving force. Using equation (3) in the manuscript, the average ΔP for CO₂ will be ca. 3 bar giving a flux of ca. 0.01 mol-m²-s and a larger required surface area than the one stated.

I do not think that the focus of this paper is to claim potential commercial success. I continue to support its publication based on the scientific merit. But if the authors would like to make comments regarding process, detailed calculations and all assumptions should be given in the paper so one can easily see the calculations.

Point-by-point Response to Reviewers' Comments

We thank the reviewers for their critical evaluation and constructive suggestions. Below we address the itemized comments and provide our response with the updated text, figures, tables and relevant literature references.

The reviewers' comments are given in italics, our response in normal font.

The main text is given in blue font, while changes are indicated by italics and blue font.

Reviewer 1:

Generally, the authors have addressed my comments in the last round. Some further suggestions are as follows.

Response

Thanks. The following are our responses to the points raised by the reviewer. We hope the reviewer will be satisfied with the responses.

Comment 1

In my view, the promising preliminary results of MFI and SSZ-13 zeolite membranes should be presented in the supporting info of this submission, because it is vitally important to demonstrate the general applicability of this new method.

Response

As suggested, the preliminary results of MFI and SSZ-13 zeolite membrane were presented in the supporting information as Supplementary Table S4. Meanwhile, the main text was revised as well: "The TMCT approach was also feasible to MFI and SSZ-13 zeolite membranes (Supplementary Table S4)." For a quick view, the Supplementary Table S4 was shown below.

Supplementary Table S4 Separation performance of MFI and SSZ-13 zeolite membranes detemplated by convention calcination (CC) and TMCT method.

No	Membrane	Detemplation	Separation performance		
			Feed	$P_1 / \text{mol m}^{-2} \text{s}^{-1} \text{Pa}^{-1}$	α
M1	MFI	CC450 ^c	PX/OX ^a	2.6×10^{-8}	11
M2	MFI	CC450 ^c	PX/OX ^a	2.1×10^{-8}	20
M3	MFI	TMCT ^d	PX/OX ^a	2.6×10^{-8}	82
M4	MFI	TMCT ^d	PX/OX ^a	4.7×10^{-8}	45

M5	SSZ-13	CC450 ^c	CO ₂ /CH ₄ ^b	2.4×10 ⁻⁷	77
M6	SSZ-13	CC450 ^c	CO ₂ /CH ₄ ^b	2.6×10 ⁻⁷	81
M7	SSZ-13	TMCT ^d	CO ₂ /CH ₄ ^b	3.4×10 ⁻⁸	11
		TMCT ^d +CC450 ^c	CO ₂ /CH ₄ ^b	2.0×10 ⁻⁷	41
M8	SSZ-13	TMCT ^d	CO ₂ /CH ₄ ^b	5.1×10 ⁻⁸	22
		TMCT ^d +CC450 ^c	CO ₂ /CH ₄ ^b	2.4×10 ⁻⁷	100

^a: both components had same partial pressure of 2 kPa at 150 °C; ^b: equimolar CO₂/CH₄ binary mixture at 25 °C and feed pressure of 2 bara; ^c: calcination at 450 °C for 12 h with heating and cooling rate of 0.5 °C min⁻¹; ^d: TMCT at 700 °C for 1 min.

Comment 2

As for comparing with other reported inorganic membranes made of zeolites, MOF, and carbons in terms of both selectivity and permeability, the authors are suggested to list best-performers in the literature for a fair comparison.

Response

Thanks for your good suggestion. As suggested, the separation results of best-performance inorganic membranes (zeolites, carbon molecular sieves and MOFs) were added in the updated Fig 5c for comparison. The corresponding references (Refs. 36, 37, 39, 40, 44) were added and the main text was revised as: “Our DD3R zeolite membranes were the most CO₂-selective compared to other zeolite membranes (e.g., SAPO-34³², SSZ-13³³, AIPO-18³⁴), carbon molecular sieve membranes^{35, 36, 37} and MOF membranes (e.g., ZIF-8³⁸, MOF-1³⁹, ZIF-62⁴⁰) in spite of their moderate permeabilities.”

Fig. 5 TMCT procedure optimization and CO₂/CH₄ separation performance. (c) Comparison on CO₂/CH₄ separation performance, involving commercial cellulose acetate (CA) membranes³¹, polyimide membranes²⁹, Matrimid membranes³⁰, carbon molecular sieve (CMS) membranes^{35, 36, 37}, MOF membranes^{38, 39, 40} and other 8MR zeolite membranes^{28, 32, 33, 34, 44}.

Reference

- 36 Hou M, et al. Carbon molecular sieve membrane with tunable microstructure for CO₂ separation: Effect of multiscale structures of polyimide precursors. *J Membr Sci* **635**, 119541 (2021).
- 37. Qiu W, Leisen JE, Liu Z, Quan W, Koros WJ. Key features of polyimide-derived carbon molecular sieves. *Angew Chem Int Ed* **60**, 22322-22331 (2021).
- 39. Rui Z, James JB, Kasik A, Lin YS. Metal-organic framework membrane process for high purity CO₂ production. *AIChE J* **62**, 3836-3841 (2016).
- 40. Wang Y, et al. A MOF glass membrane for gas separation. *Angew Chem Int Ed* **59**, 4365-4369 (2020).
- 44 Tang X, et al. Fast synthesis of thin SSZ-13 membranes by a hot-dipping method. *J Membr Sci* **629**, 119297 (2021).

Comment 3

As the authors acknowledge the limitation of XRD, they are advised to revisit their elucidation of the mechanism accordingly.

Response

Thanks. Due to the limitation of XRD, it is hard to make the characterization condition to be consistent with the instantaneous overheating process at 700 °C (TMCT). For this reason, we employed the *in-situ* XRD to compare the thermal expansion behaviors of three samples, including fresh, TMCT-treated and detemplated DD3R zeolite, using the same temperature program from 25 to 900 °C in air atmosphere. We concerned more about the variation in crystal size of the DD3R zeolite at the temperature less than 510 °C because of no decomposition of the template occluded in the zeolitic cages (See the TGA results in Supplementary Figure S8). As mentioned in the section, the non-selective macro-pores were generated for the fresh membrane sintering prior to 510 °C (Fig. 4g). Therefore, the *in-situ* XRD results were sufficient to support the elucidation of the mechanism.

To make it clear, the main text was revised as: “The thermal expansion was further tracked by in-situ X-ray diffraction, wherein the DD3R zeolite powders were subject to heating from room temperature to 900 °C step by step with 5 minutes equilibrium time prior to data collection. The diffraction would be irrelevant with the ramp rate and duration time prior to 510 °C since the templates were intact at the temperatures (Supplementary Figure S8).”

Reviewer 2:

I do not agree with the reply of the authors on my comment regarding flux. The $0.1 \text{ mol m}^{-2} \text{ s}^{-1}$ flux that the authors mention is for pure CO_2 feed at 31 bar with small stage cut. In the actual process, even if we consider 55 bar with 30% CO_2 feed and 1 bar permeate (the second case in the reply of the authors), one has to account for the retentate being only 2% in CO_2 . At this large stage cut, the membrane will operate eventually at low driving force. Using equation (3) in the manuscript, the average ΔP for CO_2 will be ca. 3 bar giving a flux of ca. $0.01 \text{ mol m}^{-2} \text{ s}^{-1}$ and a larger required surface area than the one stated.

I do not think that the focus of this paper is to claim potential commercial success. I continue to support its publication based on the scientific merit. But if the authors would like to make comments regarding process, detailed calculations and all assumptions should be given in the paper so one can easily see the calculations.

Response

Thanks. We agree with the reviewer's comment. As suggested by the reviewer, we revised the relevant descriptions and removed the commercial discussions in the text. The potential commercial application of the membranes will be carefully evaluated in the further independent work. The main text was revised as follow:

“Ultra-selective hollow fiber DD3R zeolite membranes were scaled up, which are appealing to CO₂-rich natural gas upgrading.” ... “It can be anticipated that the hollow fiber DD3R zeolite membranes would be of great interest in CO₂-rich natural gas upgrading owing to their virtue of high selectivity.”... “The TMCT approach overcame the barrier for preparation of ultra-selective DD3R zeolite membrane and paves the way to CO₂-rich natural gas upgrading based on the zeolite membranes.”

REVIEWERS' COMMENTS

Reviewer #1 (Remarks to the Author):

My questions and comments have been well addressed by the authors. Thus, I recommend to accept the manuscript for publication.

Reviewer #2 (Remarks to the Author):

The authors have addressed all my comments and I am pleased to recommend acceptance of this manuscript. Congratulations to the authors for a nice work.

Point-by-point Response to Reviewers' Comments

We thank the reviewers for their critical evaluation and constructive suggestions.

The reviewers' comments are given in italics, our response in normal font.

Reviewer 1:

My questions and comments have been well addressed by the authors. Thus, I recommend to accept the manuscript for publication.

Response

Thanks. We are happy that the reviewer is satisfied with the responses.

Reviewer 2:

The authors have addressed all my comments and I am pleased to recommend acceptance of this manuscript. Congratulations to the authors for a nice work.

Response

Thanks again for the reviewer's valuable comments.